# The Immunohistochemical Prognostic Value of Nuclear and Cytoplasmic Silent Information Regulator 1 Protein Expression in Saudi Patients with Breast Cancer

**DOI:** 10.3390/biom15010050

**Published:** 2025-01-02

**Authors:** Bayan Alharbi, Alia Aldahlawi, Mourad Assidi, Fatemah Basingab, Kawther Zaher, Jehan Alrahimi, Sara Mokhtar, Jaudah Al-Maghrabi, Abdelbaset Buhmeida, Kaltoom Al-Sakkaf

**Affiliations:** 1Department of Medical Laboratory Sciences, Faculty of Applied Medical Sciences, King Abdulaziz University, Jeddah 21589, Saudi Arabiaamokhtar@kau.edu.sa (S.M.); 2Laboratory, King Salman Medical City, Madinah 42319, Saudi Arabia; 3Department of Biological Sciences, Faculty of Sciences, King Abdulaziz University, Jeddah 21589, Saudi Arabiafbaseqab@kau.edu.sa (F.B.);; 4Immunology Unit, King Fahad for Medical Research, King Abdul-Aziz University, Jeddah 21589, Saudi Arabia; 5Institute of Genomic Medicine Sciences, King Abdulaziz University, Jeddah 21589, Saudi Arabia; 6Department of Pathology, Faculty of Medicine, King Abdulaziz University, Jeddah 21589, Saudi Arabia; jalmaghrabi@hotmail.com; 7Department of Pathology, King Faisal Specialist Hospital and Research Center, Jeddah 23433, Saudi Arabia

**Keywords:** silent information regulator 1 (SIRT1), breast cancer (BC), BC molecular subtypes, immunohistochemistry, tissue microarray

## Abstract

Background: The mammalian NAD-dependent deacetylase sirtuin-1 family (named also silent information regulator or SIRT family, where NAD stands for “nicotinamide adenine dinucleotide” (NAD)) appears to have a dual role in several human cancers by modulating cell proliferation and death. This study examines how SIRT1 protein levels correlate with clinicopathological characteristics and survival outcomes in patients with breast cancer. Methods: A total of 407 BC formalin-fixed paraffin-embedded (FFPE) samples were collected from King Abdulaziz University Hospital, Saudi Arabia. SIRT1 was stained on tissue microarray slides using automated immunohistochemistry. Results: All BC subtypes expressed more nuclear SIRT1 proteins than their cytoplasm counterparts. In luminal A, luminal B, and TNBC, nuclear and cytoplasmic SIRT1 were highly associated (*p* < 0.001). Kaplan–Meier analysis showed reduced disease-specific survival (DSS) in H2BC with high SIRT1 nuclear expression (*p* = 0.001, log-rank). Moreover, the cytoplasmic expression of SIRT1 in HER2-positive BC was associated with a larger tumor size (*p* = 0.036) and lymph node metastasis (*p* = 0.045). Nuclear SIRT1 expression was also positively associated with lymph node metastasis (LNM) (*p* = 0.048). As low-grade tumors had a higher frequency of SIRT1 protein expression than other groups, SIRT1 expression was associated with a favorable prognosis in patients with luminal A BC (*p* < 0.001). Conclusions: SIRT1 expression seems to be involved in different molecular pathways either suppressing or promoting tumor growth depending on the subtype of BC. These molecular functions require further investigations and validation on larger BC cohorts.

## 1. Introduction

Breast cancer is a growing global threat to human health and is ranked first among the types of malignancies related to female cancers in the vast majority of countries [1]. BC and its treatments affect women’s quality of life in multiple physical and psychological domains [2]. Despite the molecular etiology of cancer initiation and progression not being fully understood, it is an established fact that cancer incidence steadily increases with age. Normal aging causes accumulating mutations and sometimes induces changes in the tissue microenvironment, which are more propitious for the growth of malignant cells [3]. In this regard, genes that regulate longevity and are specifically involved in nicotinamide adenine dinucleotide (NAD)-dependent protein deacetylation, such as the silent information regulator 2 (SIR2) gene, are suggested to indicate a potential mechanistic link between the cellular functions determining aging and carcinogenesis [4]. In mammals, there are seven SIRT-independent genes/proteins (SIRT1–7) that differ in their cellular functions, substrate specificities, and subcellular localization [5,6,7]. SIRT1 is the most well-studied member of this family. SIRT1 has been found to be located primarily in the cell nucleus and could also be found in the cytoplasm upon oxidative stress [8]. The biological functions of SIRT1 are mostly demonstrated through its deacetylation activity of a growing list of non-histone substrates [9]. Intriguingly, SIRT1 appears to have a dichotomous role in cancer as an inhibitor and/or promoter. On the one hand, SIRT1 protects organisms against cancer by maintaining genomic integrity and DNA repair [10,11]. Also, the link between SIRT1 protein loss and tumor emergence has been evidenced by several models, where SIRTs knockout mice are more prone to tumor formation [12,13,14,15]. However, some established tumors tend to require enzymes to promote cell survival and proliferation under oxidative stress by allowing unchecked cell division and inhibiting senescence, which are directly involved in tumorigenesis [16,17,18]. The plausible explanation of this dual aspect of SIRT1 in tumor cells could be related to its modulation of different pivotal cellular pathways and biological processes, such as genome integrity, cell growth, cell cycle, and cell death, which indicates that this molecule seems to be tightly regulated in carcinogenesis [19,20]. Therefore, this study aims to assess the expression of the SIRT1 protein in BC tissue according to its subcellular location in a more widespread panel of cases, using immunohistochemistry (IHC) and tissue microarrays (TMAs). The molecular classification of BC is an important aspect of clarifying differences in biological features between subgroups, which allows for individualized treatment through targeting a pathway and/or factor associated with BCs of each subtype [21,22]. To the best of our knowledge, this is the first study that analyzes the association between the SIRT1 protein expression and the clinicopathological features of patients with breast cancer in Saudi Arabia, exploring itss potential prognostic value.

## 2. Materials and Methods

### 2.1. Patients and Samples

This was a retrospective study, which included 407 tissue samples from patients diagnosed with BC with a median follow-up period of 60.92 (±3.7) months and the corresponding 360 lymph node (LN) tissues of these patients. The formalin-fixed paraffin-embedded (FFPE) tissue samples and patient data with their clinicopathological information, including age, histotype, tumor size, tumor grade (Scarff–Bloom–Richardson grading system (SBR)), LN (lymph node) status, VI (vascular invasion), tumor margin, ER status, PR status, and HER2 status, were retrieved from the pathology lab of King Abdulaziz University Hospital (KAUH). This work was granted ethical approval with number 012-CEGMR-ETH from the Center of Excellence in Genome Medicine Research (CEGMR)’s ethical committee. This study adhered to the guidelines set by the ethics committee of the KAUH, Jeddah, Saudi Arabia.

### 2.2. Tissue Microarray Construction

FFPE BC samples were utilized for TMA. The histological representativeness for each case was verified by an experienced pathologist. The area of interest on the hematoxylin–eosin (H&E)-stained slide of the BC donor block was identified and marked. Using a tissue microarrayer (3D HISTECH TMA MASTER Tissue Microarrayer, FL, USA), the selected area was cored twice with a 0.6 mm diameter cylinder and mounted to a recipient paraffin block. Typical human placenta tissue FFPE cores were included at specific positions on the same slide, along with the patients’ tissues, as a positive control and for slide orientation purposes. Once the master TMA recipient blocks were ready, they were subjected to microtome for cutting TMA slides at a thickness of 4 micrometers, then transferred to a positively charged slide for IHC staining. The sample positions were specifically assigned in terms of the X- and Y-coordinates, which were recorded on Microsoft Excel table sheets.

### 2.3. Immunohistochemistry Assay

#### 2.3.1. Immunohistochemical Staining

SIRT1 protein expression was investigated by immunohistochemistry (IHC) staining using an anti-SIRT1 antibody (catalog # ab110304, clone number #19A7AB4, Abcam, Cambridge, United Kingdom) at 1:50 dilution. Protein detection was performed using the Benchmark XT, Ventana Medical System, Inc. (Tucson, AZ, USA), and the ULTRAVIEW TM DAB visualizing system, which is an indirect, biotin-free system. The slides were labeled by a barcode, which corresponded to the protocol to be performed. All steps of staining were carried out using an automated close-system protocol, except for the addition of the primary Ab, which was applied manually. The automated Ventana protocol processed the slides by deparaffinizing the paraffin-embedded tissue sections using EZ Prep™ (Sigma–Aldrich, St. Louis, MO, USA, .Cat. # 5279771001; Lot# F23209). The antigen retrieval procedure then took place using a tris-based buffer called the cell-conditioning buffer (CC1) (cat. # 5279801001; lot# F30258) (short: 8 min conditioning; and mild: 36 min conditioning for SIRT1). Afterward, 50 μL of the diluted Ab was added, followed by the steps of the UltraView Universal DAB Detection Kit (Cat.# 5269806001; Lot# F30427), which included a DAB inhibitor containing 3% H_2_O_2_, HRP multimer (<50 μg/mL), DAB chromogen (0.2% DAB), DAB H_2_O_2_ (0.04% H_2_O_2_), and DAB copper (CuSO_4_ 5 g/L). At the end of each incubation step, the sections were washed to remove unbound material. Also, after each staining reagent, a liquid coverslip (LCS) was applied to minimize the evaporation of the aqueous reagents from the slide. The final stage of staining consisted of counterstaining with hematoxylin II (cat.# 5266726001; lot# F29048) for 8 min and post-counterstaining with blue reagent (cat.# 5266769001; lot# F26734) for 4 min. The slides were removed from the automated stainer, washed in a mild detergent, and rinsed in tap water to remove any residual buffer and detergent, dehydrated by successive immersions in different concentrations of alcohol buffer (Honeywell, Muskegon, MI, USA; catalog # 32221) (70, 95, and 100%), and then cleared in xylene (Sigma–Aldrich, St. Louis, MO, USA; catalog # 33817) for 3 min; this was carried out twice for each solution. Finally, one drop of the mounting medium was applied onto the slide and covered by a glass coverslip.

#### 2.3.2. Evaluation of SIRT1 Staining

SIRT1 staining was evaluated and scored manually by two qualified pathologists blinded to the clinicopathological parameters of the patients. The nuclear and cytoplasmic immunoreactivity were scored separately in a semi-quantitative manner by assessing the intensity of staining in different areas of the section. The staining intensity assessment was divided into four categories: negative (0), weakly positive (1+), moderately positive (2+), and strongly positive (3+). The Olympus CH20 binocular light microscope (Tokyo, Japan) was used at ×10 magnification for scanning to obtain an overall impression of the staining and then at ×40 magnification to estimate the fractions (f0, f1, f2, and f3). The staining intensity of individual cells and the fraction of positively stained cells were used to calculate the staining index (I) score, which ranged from 0 to a maximum of 300 [23], using the following formula:I = (0 × f0) + (1 × f1) + (2 × f2) + (3 × f3)

The scores of SIRT1 expression/staining intensity were defined by the overall total index score I. This method allows for further flexibility in selecting suitable cutoffs at both the nuclear and cytoplasmic levels. The staining expression was regarded as SIRT1− (low SIRT1 staining/expression; index score < mean) and SIRT1+ (high staining/expression; index score > mean).

### 2.4. Statistical Analysis

Statistical analyses were performed using SPSS software packages (version 23; IBM, New York, NY, USA). SIRT1 protein expression in tissues of BC subtypes was compared using a Kruskal–Wallis test. Spearman’s rank correlation was calculated to assess the correlation between nuclear SIRT1 expression and cytoplasmic SIRT1 expression. The scores of SIRT1 expression by IHC in BC were analyzed by the Chi-square test followed by Bonferroni’s post hoc test to determine any statistically significant data between variables. Logistic regression and multivariate logistic regression analyses were performed to evaluate risk factors and identify factors that were independently prognostic of the risk. Univariate survival analysis using the Kaplan–Meier method was performed for evaluating both disease-free survival (DFS) and disease-specific survival (DSS). DSS was defined as the time from diagnosis to death (due to disease) or to the date on which a patient had last been seen alive. For DFS, it was defined as the time from diagnosis to the appearance of recurrent disease or the date on which a patient had last been seen disease-free, respectively. Patients who died of other or unknown causes were excluded from the survival analysis. All statistical tests were two-sided, tests with *p*  ≤  0.05 were considered statistically significant, and *p*  ≤  0.01 was viewed as highly statistically significant.

## 3. Results

### 3.1. The Correlation of BC Molecular Subtypes with the Clinicopathological Features

The clinicopathological variables of patients with BC are described in Table 1. This study reported 407 female patients with BC classified into four molecular subtypes. The classification was based on the IHC subtyping system of BC described by Sung et al. [24]: 178 (44%) luminal A (ER+ and/or PR+, Her2−), 90 (22%) luminal B (ER+ and/or PR+, Her2+), 62 (15%) H2BC (Her2-enriched; (ER−, PR−, Her2+)), and 77 (19%) TN (triple-negative; (ER−, PR−, Her2−)). Half of the patients (50.2%) were above the age of 50 years. Tumors were graded according to the modified Scarff–Bloom–Richardson grading system (SBR), and most of the patients presented with grade II.

Table 1 presents the correlation between different breast cancer (BC) molecular subtypes and various critical clinicopathological features. Interestingly, the BC molecular subtypes demonstrated a remarkably strong and highly significant correlation with several key parameters, including patient age (*p* = 0.028), tumor vascular invasion (*p* = 0.001), tumor size (*p* = 0.006), and tumor grade (*p* = 0.001). These findings highlight the substantial influence of molecular subtypes on the biological behavior and clinical presentation of BC. However, the analysis revealed no statistically significant correlation between BC molecular subtypes and certain other clinicopathological characteristics, such as tumor histotype (*p* = 0.245), lymph node status (*p* = 0.249), and tumor margin (*p* = 0.440). This suggests that, while molecular subtypes are closely associated with specific aggressive tumor features, they may not play a direct role in influencing these particular attributes. The observed associations underscore the complexity and heterogeneity of BC, emphasizing the importance of molecular subtyping in understanding tumor biology. These correlations could have implications for tailoring treatment strategies, as patients with distinct molecular subtypes may exhibit different responses to therapy or prognostic outcomes based on the associated clinicopathological features. Further research into these relationships may provide deeper insights into the mechanisms linking molecular subtypes to tumor behavior, ultimately contributing to more personalized and effective approaches to BC management.

### 3.2. Protein Expression Status of Nuclear and Cytoplasmic SIRT1 in BC Subtypes

SIRT1 immunostaining in the 17 coated microarray slides of BC tissues showed that this protein was expressed in both the nucleus and cytoplasm, with varying intensities, as shown in Figure 1. There was no significant difference in SIRT1 protein expression between the various subtypes in the nucleus (*p* = 0.089) or cytoplasm (*p* = 0.995) (Figure 2). However, the results showed that the protein level of nuclear SIRT1 was higher in BC tissue compared to the cytoplasm in all other BC subtypes included in this study. Furthermore, the protein expression levels of SIRT1 in the eight TMA slides for LN were more abundant in the nucleus than in the cytoplasm, and primary tumors exhibited higher levels of SIRT1 compared to patients with LNM (Table 2). The results also showed that there was a significant positive association between nuclear and cytoplasmic SIRT1 in luminal A BC (r2 (176) = 0.36, *p* < 0.001), luminal B BC (r2 (88) = 0.397, *p* < 0.001), and TNBC (r2 (75) = 0.44, *p* < 0.001). Nevertheless, no significant association was detected between nuclear and cytoplasmic SIRT1 in H2BC (r2 (60) = 0.049, *p* = 0.704) (Figure 3).

### 3.3. Associations Between SIRT1 Expression and BC Receptor Statuses

Nuclear SIRT1 proteins showed statistically significant associations with ER status, PR status, and Her2 status (*p* = 0.001). The overexpression of nuclear SIRT1 occurred in 49% of ER-positive tumors, while SIRT1 expression was lower in 78% of PR-positive and 66% of H2BC subtypes, respectively (Figure 4). However, cytoplasmic SIRT1 expression did not show any significant associations with ER, PR, H2BC, and TN status (*p* = 0.862), as shown in Table 3.

### 3.4. Association of SIRT1 with the Clinicopathological Characteristics of the Patients

#### 3.4.1. Nuclear SIRT1 Protein Expression in Relation to Clinicopathological Characteristics

Nuclear SIRT1 expression was examined in relation to the patients’ clinicopathological features for each intrinsic subtype. Our data showed (Table 4) that the SIRT1 protein was significantly associated with luminal A and TN BC. In luminal A, the level of expression of SIRT1 varied between different tumor grades (*p* < 0.001). Higher SIRT1 expression was observed mainly in 76% of grade I, 34% of grade II, and 50% of grade III carcinomas. These results support the notion that SIRT1 overexpression is observed mainly in low-grade tumors. However, the higher expression of SIRT1 in the TN type was associated with lymph node metastasis (*p* = 0.048) (Figure 5A). Nuclear SIRT1 protein expression was shown to be an independent predictive variable in multivariate logistic regression analyses in luminal A BC (*p* = 0.029, odds ratio [OR] of 0.313 confidence interval (CI) 0.890 (upper)–0.110 (lower)) but not in TNBC (*p* = 0.053, odds ratio [OR] of 3.130 CI 9.965 (upper)–0.983 (lower)).

#### 3.4.2. Cytoplasmic SIRT1 Protein Expression in Relation to Clinicopathological Characteristics

The association between clinicopathological factors and cytoplasmic SIRT1 in the various tissues of BC intrinsic subtypes is shown in Table 5. The expression patterns of cytoplasmic SIRT1 were significantly associated with H2BC and TNBC. The overexpression of cytoplasmic SIRT1 in Her2 types showed a significant association with tumor size (*p* = 0.036). SIRT1 positivity was observed in bigger tumors (57.1%) compared to the other groups. Also, SIRT1 protein expression in the cytoplasm was shown to be an independent predictive variable in multivariate logistic regression analysis in H2BC (*p* = 0.016, odds ratio [OR] 0.107 of CI 0.662 (upper)–0.017 (lower)). Similarly to the nuclear expression of SIRT1 in TN, cytoplasmic SIRT1 expression was associated with lymph node metastasis (*p* = 0.045), without being an independent predictive variable (*p* = 0.051, odds ratio [OR] of 3.420 CI 11.780 (upper)–0.993 (lower)) (Figure 5B).

### 3.5. The Correlation Between SIRT1 Protein Expression Patterns and Outcomes in Patients with BC

The associations between SIRT1 expression patterns and a possible prognosis value were investigated using Kaplan–Meier plot curves. This correlation is illustrated in Figure 4, emphasizing the prognostic value of SIRT1 and the significant impact of its cellular localization. Specifically, the cytoplasmic expression of SIRT1 demonstrated a more favorable prognostic association (A and B) compared to nuclear expression (C and D). Patients exhibiting high cytoplasmic SIRT1 protein expression profiles showed noticeably reduced rates of disease recurrence and death, suggesting a protective role of cytoplasmic SIRT1 in BC progression. This contrasted sharply with the outcomes observed in patients with high nuclear SIRT1 expression profiles, where a clear association with a worse prognosis was evident, characterized by higher disease recurrence rates and an increased death risk. For instance, as depicted in Figure 6D, at a 5-year follow-up, only about 10% of the patients with low nuclear SIRT1 expression had died, compared to approximately 40% of the patients with high nuclear SIRT1 expression. This significant difference (*p* < 0.049, log-rank test) highlights the adverse prognostic implications of nuclear SIRT1 localization. These findings underscore the potential utility of SIRT1 protein localization as a prognostic biomarker in BC. Understanding the differential roles of cytoplasmic and nuclear SIRT1 in tumor pathophysiology could also open avenues for targeted therapies to improve patient outcomes by modulating SIRT1 localization and activity.

## 4. Discussion

BC is a heterogeneous and multifactorial disease involving tumor genetics, biology, and the environment, which differs greatly among patients and remains a causative factor in female mortality [25]. Currently, an important goal of oncology is the development of novel specific molecular biomarkers to improve BC prognosis and replace standard “one size fits all” treatments [22,26]. Mammalian SIRT family members have gained tremendous attention in the past decade in cancer research, and numerous studies have demonstrated their direct implication in the carcinogenesis process of multiple human cancers. SIRT1, the most investigated isoform in the SIRTs family, belongs to class III histone deacetylases (HDACs) and shares a catalytic domain that bears NAD+-dependent deacetylases for its primary enzymatic activity [20,27,28,29,30]. Different pivotal cellular pathways and biological processes are modulated by SIRT1 [31]. However, the role of SIRT1 in breast carcinoma remains a subject of debate because of conflicting results due to its pleiotropic functions. Recently, most reports have suggested that the dichotomous role of SIRT1 in either suppressing tumors [32,33] and/or promoting tumors [18] depends on the target substrates [20]. Up until now, most studies investigating the associations between advanced-staged BC (including tumor invasion and metastatic progression) and SIRT1 expression were in Southeast Asian populations. Subgroup analyses revealed that ethnic origin has an impact on the role of SIRT1 expression and the clinicopathological features of cancer. Therefore, these findings may not be generalizable outside this population [34]. Furthermore, most SIRT1 investigations were either in vitro human cell line models [25,35] or involved a limited number of human-derived tissue samples [24]. Additionally, in these works, the expression of SIRT1 was assessed through at the transcriptional level and did not take into consideration the protein level [36]. The semi-quantitative reading of immunostaining results might be a reason for this discrepancy, as the cutoff values for SIRT1 expression differed a lot in some of the included reviews and studies. Therefore, these results are inconsistent on whether SIRT1 acts as a tumor promoter or suppressor role. In this study, the expression pattern of SIRT1 in tissues of intrinsic subtypes of BC and the corresponding LN tissues in patients with LN-positive lesions were evaluated using IHC of FFPE tissue samples. The main findings of this study can be concluded to be as follows: (a) The protein level of nuclear SIRT1 was higher in all BC subtype tissues compared to this protein’s cytoplasmic expression. (b) Primary tumors exhibited higher levels of SIRT1 compared to LN tissues. (c) There was a significant positive correlation between nuclear and cytoplasmic SIRT1 in luminal A, luminal B, and TNBC. (d) The nuclear protein expression of SIRT1 in patients with luminal A BC was significantly associated with low-grade tumors (*p* < 0.001), meaning that it could be considered a favorable prognostic factor in this population. Meanwhile, the expression pattern of cytoplasmic SIRT1 was a poor prognostic factor in H2BC, as it was associated with a larger tumor size (*p* < 0.036). In TNBC, nuclear and cytoplasmic SIRT1 protein overexpression was associated with LN metastasis (*p* = 0.048 and *p* = 0.045, respectively). (e) In H2BC, the group with high SIRT1 nuclear expression was found to be associated with a significantly reduced disease-specific survival (DSS) (*p* = 0.001, log-rank), whereas, in terms of disease-free survival (DFS), there were no significant differences between various expression levels in all BC subtypes.

Previous studies indicated that SIRT1 is upregulated in several types of tumors. Also, the extensively high SIRT1 protein levels in breast carcinoma have been found to be associated with tumor promotion and a worse prognosis [18,20,21,37,38,39,40,41,42,43]. However, Wang et al. reported that SIRT1 is expressed in lower levels in many cancer tissues, including breast cancer, relative to adjacent normal tissues, which provides strong evidence of SIRT1 antioncogenic functions [33]. In agreement with others, SIRT1 overexpression as a potential biomarker of poor prognosis in patients with BC is further supported by our observation in patients with H2BC, as the group with higher nuclear SIRT1 protein levels was significantly associated with a reduced DSS but not DFS. However, previous publications by Cao et al. and Wu et al. showed prognostic value for both DSS and DFS [18,41]. The first non-histone target identified for SIRT1 is p53, which is known to play an important role in cell apoptosis [38]. The association of p53 and SIRT1 was also studied by Wu and coworkers in 2012, with a sample size of 134 cases. They found an association between SIRT1 and p53 in all BC subtypes [18]. Moreover, Kuo Lin et al. confirmed that the downregulation of SIRT1 by a specific inhibitor causes an anti-breast cancer cell effect by blocking the activity of the Bcl-2 protein, which is a pro-survival protein which protects cells from apoptosis [44]. Their finding indicated that SIRT1 takes part in tumor formation by repressing genotoxic stress-induced apoptosis.

In the present study, we observed that the upregulation of nuclear SIRT1 was associated with ER-positive tumors, while PR-positive and Her2-positive tumors had low SIRT1 expression. In fact, the correlation between SIRT1 protein expression, estrogen receptor ER, and the Her2 subtype was seen using several TMA evaluations with variable cutoff points. In this regard, we hypothesized that SIRT1 most probably has a tumor suppressor role or an oncogenic role in these subtypes. Consistent with Rifaï et al., who examined SIRT1 expression patterns according to intrinsic subtypes, we found that HRBC subtypes had the highest expression of the SIRT1 protein, and the downregulation of SIRT1 was associated with increased BC aggressivity [21]. Another study confirmed this finding at the gene level, as patients with ER+ PR+ Her2- breast tumors had a higher genetic expression of SIRT1 compared to TNBC [45]. Our study demonstrated the potential role of SIRT1 as a tumor suppressor in the luminal A subtype due to the nuclear up-regulation of the protein being significantly higher in grade 1 than in grades 2 and 3. This was in agreement with a previous report, which found that the overexpression of SIRT1 in BC tissues was associated with a low tumor grade [24]. Furthermore, Chung et al. confirmed that increased SIRT1 expression in the luminal A subtype was significantly associated with a lower rate of LNM and a longer DFS [17]. In fact, patients with ER-negative BC generally tend to have the worst prognosis and are often more drug-resistant than ER-positive tumor cases [44]. Nonetheless, ER-positive BC subtypes can show the development of metastasis in some cases. This led Kim and coworkers to construct a biomarker prediction model of LNM in these subtypes using apoptosis-related proteins, including SIRT1. Their finding was consistent with the earlier studies, and they concluded that an elevated expression of SIRT1 exerts a tumor-suppressive role and was associated with a low frequency of LN metastasis in ER-positive BC subtypes [46]. However, findings presented in other reports suggest the oncogenic potential of SIRT1 in ER-positive BCs [5,47]. ER alpha and SIRT1 have been shown to cooperate in the development of mammary tumorigenesis, where SIRT1 inactivation eliminates the E2-mediated promotion of BC growth and positively triggers p53-mediated apoptosis [47].

In our study, a relation between both nuclear and cytoplasmic SIRT1 protein expression and TNBC subtypes was detected. The few studies that have investigated SIRT1 biological behaviors in TNBC show inconsistent findings. Through both in vivo and in vitro studies, Chung et al. reported that increased expression of SIRT1 in TNBC was associated with tumor invasion and LNM [17]. Notably, Sun and coworkers performed a meta-analysis study to investigate the clinicopathological significance of SIRT1 expression in 16 cancer types. They found that the elevated expression of SIRT1 could predict metastasis and advanced malignancy [34]. Nevertheless, other studies found that SIRT1 was significantly lower in the TNBC subtype at both transcriptional and translational levels, and the deacetylation of the mutant form of p53 in an SIRT1-dependent manner tended to suppress the growth of the TNBC cell line [25]. Future studies are needed to focus on the challenging group of patients with TNBC, as they do not express Her2, ER, nor PR receptors. So, an alternative route of signaling in this subtype should be detected and targeted.

Within our study, we investigated the relationship between nuclear and cytoplasmic SIRT1 expression. A positive association between SIRT1 subcellular localization was observed in luminal A, luminal B, and TNBC; in contrast, no such relationship was observed in H2BC. These results could be related to the specific regulation of SIRT1 expression or functions in molecular BC subtypes. Follow-up studies are crucial for validating and then explaining SIRT1 activities, as well as identifying an important mode of regulation of specific pathways via this enzyme.

Our results did not show abundant protein expression of SIRT1 in the cytoplasm of BC tissues. Nevertheless, we found an association between cytoplasmic SIRT1 and a prognostic factor related to migration and invasion in TNBC. Also, the overexpression of cytoplasmic SIRT1 was associated with a larger tumor size in H2BC, which might indicate the ability of SIRT1 to regulate biological behaviors, including colony formation and cell cycle progression in BC cells. Byles et al. demonstrated that SIRT1 is mainly located in the nucleus of normal cells, while an aberrant cytoplasm localization of SIRT1 greatly contributes to its cancer-specific function by targeting cytoplasmic proteins [48]. This finding is consistent with previous studies on several cytosolic targets that can be deacetylated and activated by SIRT1, including key EMT markers such as E-cadherin and cancer growth signaling ones such as IGF-1 and PI3K/Akt [5,6,7]. At present, only two studies have suggested that there exists a close relationship between cytoplasmic SIRT1 and the progression of ovarian carcinoma [49] and squamous cell carcinoma (SCC) [50]. This is a novel finding that could contribute to confirming this theory in BC tissues, as the available data only concern nuclear SIRT1 levels, neglecting its cytoplasmic expression. Limitations of this study are the lack of information on the numerical or TNM staging of cancer, which could help in understanding how aggressive the cancer observed is. Another limitation is the impact of missing data on the conclusions that can be drawn.

## 5. Conclusions

SIRT1 has a controversial role in carcinogenesis, with strong evidence regarding its involvement in tumors, indicating potential oncogenic and/or suppressive roles. Our data suggest that SIRT1 may have a dual role in BC depending on the molecular subtypes. The overexpression of SIRT1 could serve as a valuable prognostic marker for assessing BC status, as it is associated with poor outcomes. It is also possible that the association between higher SIRT1 expression and favorable prognostic factors may contribute to this protein’s antioncogenic function. Further mechanistic studies of epigenetic abnormalities induced by SIRT1 are needed, which may reveal new avenues for the development of more effective prognostic and therapeutic agents.

## Figures and Tables

**Figure 1 biomolecules-15-00050-f001:**
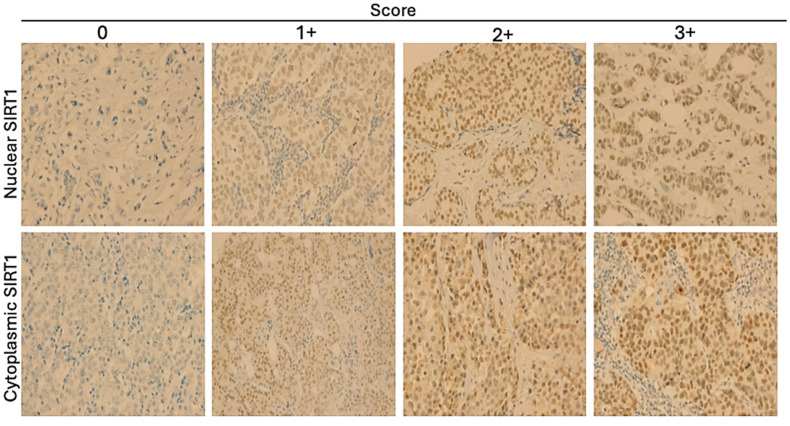
Representative cases of IHC staining of SIRT1 ranging from no expression (0) to high expression (3+) on BC samples (10× magnification). The first row shows nuclear SIRT1 expression, and the second row shows cytoplasmic SIRT1 expression.

**Figure 2 biomolecules-15-00050-f002:**
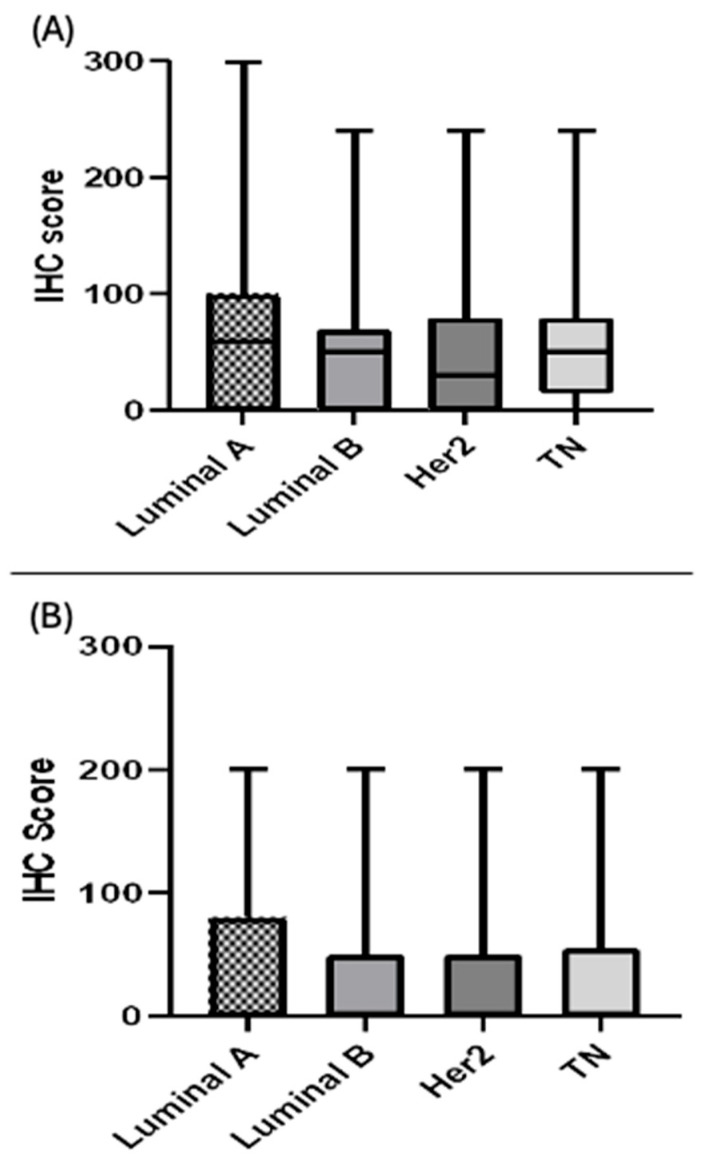
Semi-quantitative scoring of SIRT1 immunohistochemistry in the nucleus and cytoplasm of different BC subtypes: (**A**) nuclear SIRT1 protein expression and (**B**) cytoplasmic SIRT1 protein expression.

**Figure 3 biomolecules-15-00050-f003:**
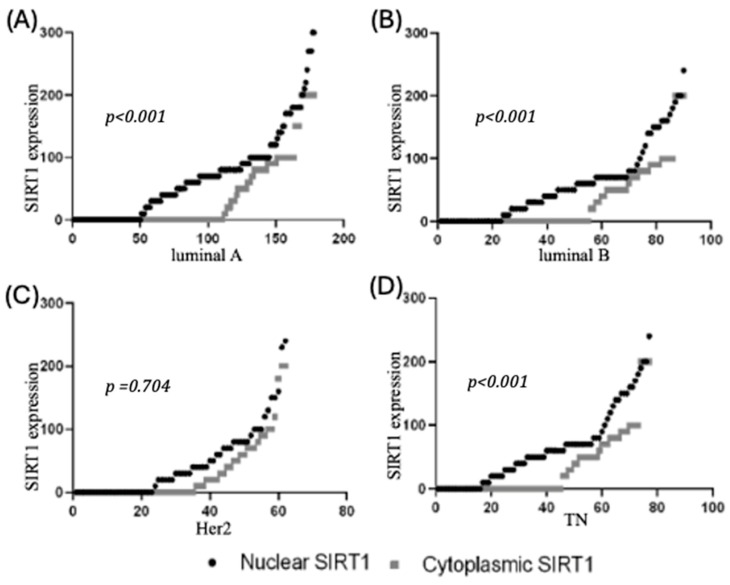
Correlation between nuclear and cytoplasmic SIRT1 in BC subtypes. Positive correlations were found only in luminal A, luminal B, and TNBC BC subtypes. (**A**): nuclear and cytoplasmic SIRT1 in luminal A; (**B**): nuclear and cytoplasmic SIRT1 in luminal B; (**C**): nuclear and cytoplasmic SIRT1 in Her2; and (**D**): nuclear and cytoplasmic SIRT1 in TN.

**Figure 4 biomolecules-15-00050-f004:**
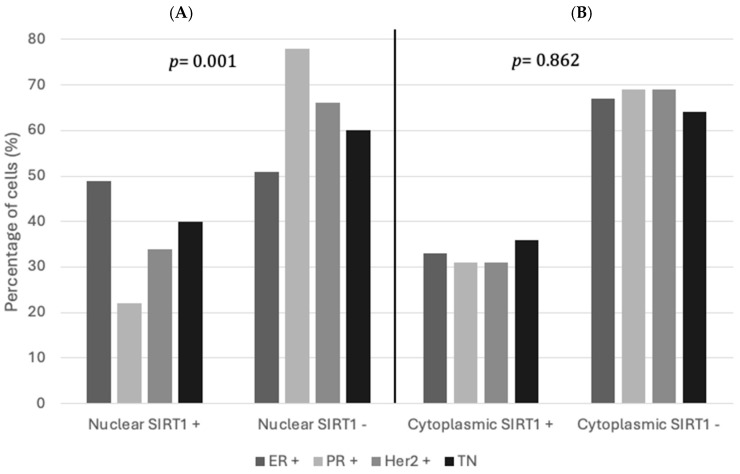
Distribution of nuclear and cytoplasmic SIRT1 expression in relation to receptor status of BC: (**A**) nuclear SIRT1 protein expression and (**B**) cytoplasmic SIRT1 protein expression. (SIRT−): low expression (<mean); and (SIRT+): high expression (>mean).

**Figure 5 biomolecules-15-00050-f005:**
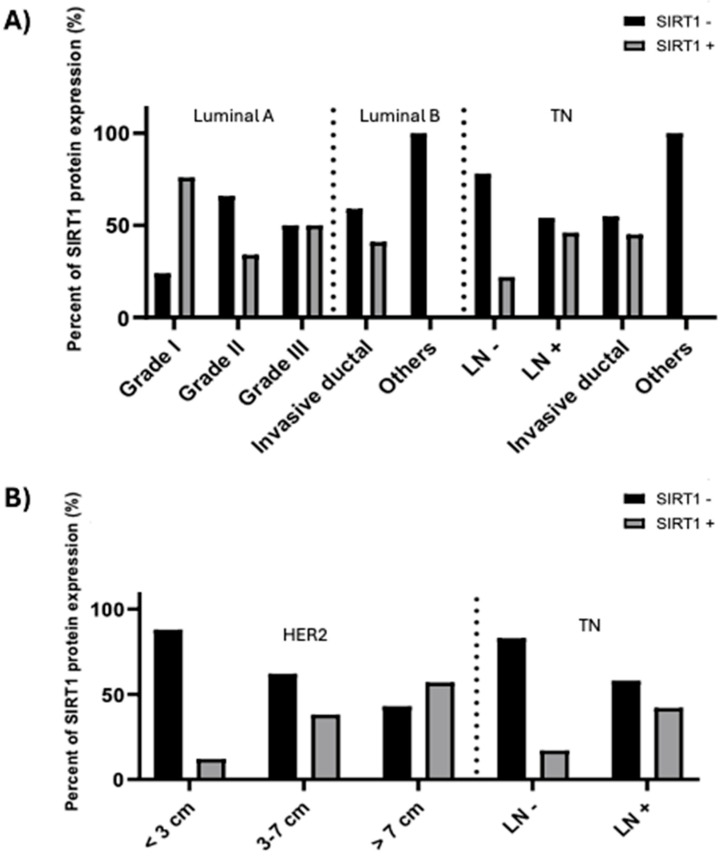
Descriptive distribution of nuclear (**A**) and cytoplasmic (**B**) SIRT1 protein expression in relation to the clinicopathological characteristics shown to be significantly associated in Table 1.

**Figure 6 biomolecules-15-00050-f006:**
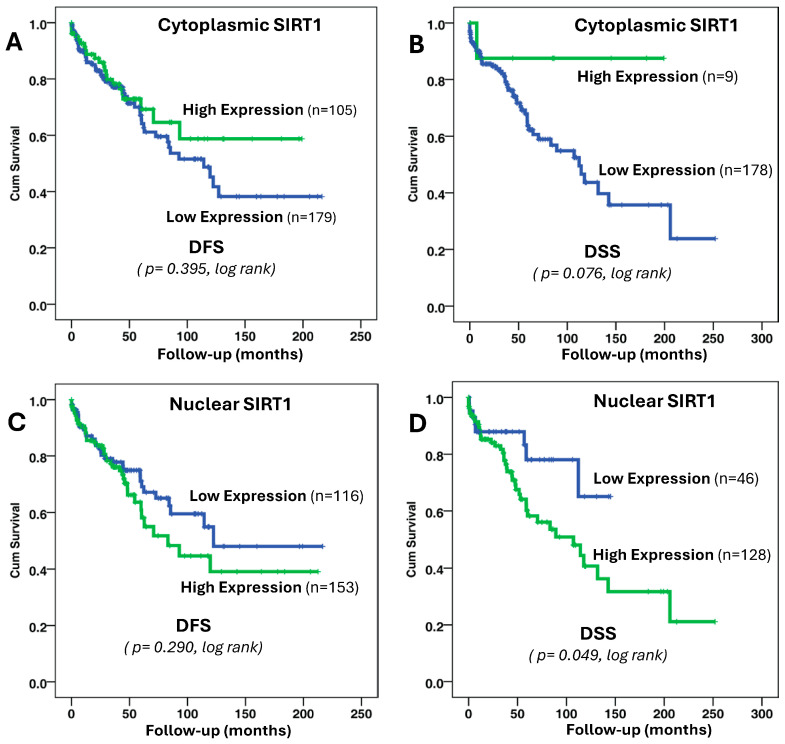
Kaplan–Meier survival curves showing both disease-free survival (DFS) and disease-specific survival (DSS) for cytoplasmic (**A**,**B**) and nuclear (**C**,**D**) SIRT1 protein expression.

**Table 1 biomolecules-15-00050-t001:** Clinicopathological features of the BC cases enrolled in this study.

Characteristics	Luminal A	Luminal B	H2BC	TNBC	*p*-Value
Patients, *n* (%)	178 (44%)	90 (22%)	62 (15%)	77 (19%)	
Histotype					0.245
Invasive ductal	153(86%)	81 (91%)	58 (95%)	69 (91%)	
Other	24 (14%)	8 (9%)	3 (5%)	7 (9%)	
Missing	1	1	1	1	
Age					0.028 *
≤50	95 (54%)	39 (43%)	38 (61%)	30 (39%)	
>50	82 (46%)	51 (57%)	24 (39%)	46 (61%)	
Missing	1	0	0	0	
Lymph Node					0.249
Negative	64 (39%)	26 (35%)	11 (23%)	23 (35%)	
Positive	101(61%)	48 (65%)	37 (77%)	43 (65%)	
Missing	13	16	14	11	
Vascular Invasion					<0.001 **
Negative	97 (67%)	31 (51%)	15 (32%)	26 (53%)	
Positive	48 (33%)	30 (49%)	32 (68%)	23 (47%)	
Missing	33	29	15	28	
Tumor Margin					0.440
Negative	151 (84%)	67 (82%)	47 (82%)	58 (83%)	
Positive	20 (16%)	15 (18%)	10 (18%)	12 (17%)	
Missing	7	8	5	7	
Tumor Size					0.006 **
<3 cm	76 (46%)	35 (42%)	16 (31%)	28 (40%)	
3–7 cm	74 (45%)	42 (51%)	21 (41%)	36 (51%)	
>7 cm	14 (9%)	6 (7%)	14 (27%)	6 (9%)	
Missing	14	7	11	7	
Tumor Grade					<0.001 **
G1	42 (27%)	14 (18%)	1 (2%)	14 (21%)	
G2	88 (56%)	49 (63%)	26 (47%)	40 (59%)	
G3	26 (17%)	15 (19%)	28 (51%)	14 (21%)	
Missing	22	12	7	9	

H2BC: Her2-enriched; TNBC: triple-negative; * significant at *p* ≤ 0.05; and ** significant at *p* ≤ 0.01.

**Table 2 biomolecules-15-00050-t002:** Expression patterns of SIRT1 within the subtypes of breast cancer tissues and the corresponding LN tissue in patients with LN-positive lesions, using the mean as a cutoff point.

Tissue Samples	Nuclear SIRT1 IHC		*p*-Value	Cytoplasmic SIRT1 IHC		*p*-Value
	Low SIRT1 Expression (≤Mean)	High SIRT1 Expression (>Mean)		Low SIRT1 Expression (≤Mean)	High SIRT1 Expression (>Mean)	
Luminal A (178 cases)	93 (52%)	85 (48%)	0.001 **	120 (67%)	58 (33%)	0.862
Luminal B (90 cases)	57 (63%)	33 (37%)		59 (66%)	31 (34%)	
H2BC (62 cases)	43 (69%)	19 (31%)		44 (71%)	18 (29%)	
TN (77 cases)	46 (60%)	31 (40%)		49 (64%)	28 (36%)	
LN tissues (360 cases)	258 (72%)	102 (28%)		316 (88%)	44 (12%)	

H2BC: Her2-enriched; TN: triple-negative; LN: lymph node; and ** significant at *p* ≤ 0.01.

**Table 3 biomolecules-15-00050-t003:** Distribution of SIRT1-positive and -negative protein expression for patients with breast carcinoma in relation to receptor status.

Receptor Status	Total	High Nuclear SIRT1 Expression (%)	Low Nuclear SIRT1 Expression (%)	High Cytoplasmic SIRT1 Expression (%)	Low Cytoplasmic SIRT1 Expression (%)
ER+	222	108 (49%)	114 (51%)	74 (33%)	148 (67%)
PR+	49	11 (22%)	38 (78%)	15 (31%)	34 (69%)
Her2+	159	54 (34%)	105 (66%)	49 (31%)	110 (69%)
TN	77	31 (40%)	46 (60%)	28 (36%)	49 (64%)

TN: triple-negative.

**Table 4 biomolecules-15-00050-t004:** Association between nuclear SIRT1 (using the mean as a cutoff point) and patients’ clinicopathological characteristics.

Patients Features	Luminal A		*p*	Luminal B		*p*	H2BC		*p*	TNBC		*p*
	SIRT1+	SIRT1−		SIRT1+	SIRT1−		SIRT1+	SIRT1−		SIRT1+	SIRT1−	
Age												
<50	46 (48%)	49 (52%)	0.308	25 (64%)	14 (36%)	0.895	26 (68%)	12 (32%)	0.841	19 (63%)	11 (37%)	0.608
>50	46(56%)	36 (44%)		32 (63%)	19 (37%)		17 (71%)	7 (29%)		27 (57%)	20 (43%)	
Lymph Node Status												
Negative	36 (56%)	28 (44%)	0.398	19 (73%)	7 (27%)	0.276	8 (73%)	3 (27%)	0.875	18 (78%)	5 (22%)	0.048 *
Positive	50 (49%)	51 (51%)		29 (60%)	19 (40%)		26 (70.3%)	11 (29.7%)		23 (53%)	20 (47%)	
Vascular Invasion												
Negative	51 (53%)	46 (47%)	0.857	18 (58%)	13 (42%)	0.674	11 (73%)	4 (27%)	0.597	15 (58%)	11 (42%)	0.698
Positive	26 (54%)	22 (46%)		19 (63%)	11 (37%)		21 (66%)	11 (34%)		12 (52%)	11 (48%)	
Tumor Margin												
Negative	75 (50%)	76 (50%)	0.197	40 (60%)	27 (40%)	0.617	34 (72%)	13 (28%)	0.439	33 (57%)	25 (43%)	0.927
Positive	13 (65%)	7 (35%)		10 (67%)	5 (33%)		6 (60%)	4 (40%)		7 (58%)	5 (42%)	
Tumor Size												
<3 cm	40 (53%)	36 (47%)	0.163	19 (54%)	16 (46%)	0.104	8 (50%)	8 (50%)	0.332	15 (54%)	13 (46%)	0.064
3–7 cm	33 (45%)	41 (55%)		30 (71%)	12 (29%)		15 (71%)	6 (29%)		24 (67%)	12 (33%)	
>7 cm	10 (71%)	4 (29%)		2 (33%)	4 (67%)		10 (71%)	4 (29%)		1 (17%)	5 (83%)	
Tumor Grade												
Low Grade (1)	10 (24%)	32 (76%)	<0.001 **	7 (50%)	7 (50%)	0.196	1 (100%)	0 (0%)	0.724	7 (50%)	7 (50%)	0.357
Intermediate (2)	58 (66%)	30 (34%)		28 (57%)	21 (43%)		18 (69%)	8 (31%)		20 (50%)	20 (50%)	
High Grade (3)	13 (50%)	13 (50%)		12 (80%)	3 (20%)		18 (64%)	10 (36%)		10 (71%)	4 (29%)	
Histotype												
Invasive Ductal	77 (50%)	76 (50%)	0.136	48 (59%)	33 (41%)	0.023 *	40 (69%)	12 (31%)	0.933	38 (55%)	31 (45%)	0.021 *
Others	16 (67%)	8 (33%)		8 (100%)	0 (0%)		2 (67%)	1 (33%)		7 (100%)	0 (0%)	

H2BC: Her2-enriched; TNBC: triple-negative BC; SIRT−: low expression; SIRT+: high expression; * significant at *p* ≤ 0.05; and ** significant at *p* ≤ 0.01.

**Table 5 biomolecules-15-00050-t005:** Association between cytoplasmic SIRT1 (using the mean as a cutoff point) and patients’ clinicopathological characteristics.

Patients Features	Luminal A		*p*	Luminal B		*p*	H2BC		*p*	TN		*p*
	SIRT1+	SIRT1−		SIRT1+	SIRT1−		SIRT1+	SIRT1−		SIRT1+	SIRT1−	
Age												
<50	66 (69%)	29 (301%)	0.494	24 (61%)	15 (39%)	0.483	27 (71%)	11 (29%)	0.985	19 (63%)	11 (37%)	0.965
>50	53 (65%)	29 (35%)		35 (69%)	16 (31%)		17 (71%)	7 (29%)		30 (64%)	17 (36%)	
LN Status												
Negative	42 (66%)	22 (34%)	0.925	21 (81%)	5 (19%)	0.105	7 (64%)	4 (36%)	0.55	19 (83%)	4 (17%)	0.045 *
Positive	67 (66%)	34 (34%)		30 (62%)	18 (38%)		27 (73%)	10 (27%)		25 (58%)	18 (42%)	
Vascular Invasion												
Negative	68 (70%)	29 (30%)	0.725	20 (65%)	11 (35%)	0.86	11 (73%)	4 (26%)	0.465	16 (61%)	10 (39%)	0.79
Positive	35 (73%)	13 (27%)		20 (67%)	10 (33%)		20 (62%)	12 (38%)		15 (65%)	8 (35%)	
Tumor Margin												
Negative	100 (66%)	51 (39%)	0.736	43 (64%)	24 (36%)	0.855	33 (70%)	14 (30%)	0.528	36 (62%)	22 (38%)	0.764
Positive	14 (70%)	6 (30%)		10 (67%)	5 (33%)		6 (60%)	4 (40%)		8 (67%)	4 (33%)	
Tumor Size												
<3 cm	51 (67%)	25 (33%)	0.808	22 (63%)	13 (37%)	0.981	14 (88%)	2 (12%)	0.036 *	17 (61%)	11 (39%)	0.807
3–7 cm	47 (63%)	27 (37%)		27 (64%)	15 (36%)		13 (62%)	8 (38%)		23 (64%)	13 (36%)	
>7 cm	10 (71%)	4 (29%)		4 (67%)	2 (33%)		6 (43%)	8 (57%)		3 (50%)	3 (50%)	
Tumor Grade												
Low Grade (1)	24 (57%)	18 (43%)	0.29	9 (64%)	5 (36%)	0.606	1 (100%)	0 (0%)	0.081	9 (64%)	5 (36%)	0.357
Intermediate (2)	61 (69%)	27 (31%)		29 (59%)	20 (41%)		21 (81%)	5 (19%)		22 (55%)	18 (45%)	
High Grade (3)	19 (73%)	7 (27%)		11 (73%)	4 (27%)		15 (54%)	13 (46%)		10 (71%)	4 (29%)	
Histotype												
Invasive Ductal	100 (67%)	50 (33%)	0.949	51 (63%)	30 (37%)	0.165	41 (71%)	17 (29%)	0.882	42 (61%)	27 (39%)	0.194
Others	16 (67%)	8 (33%)		7 (87%)	1 (13%)		2 (67%)	1 (33%)		6 (86%)	1 (14%)	

H2BC: Her2-enriched; TN: triple-negative; SIRT−: low expression; SIRT+: high expression; * significant at *p* ≤ 0.05.

## Data Availability

The original contributions presented in this study are included in the article; further inquiries can be directed to the corresponding author.

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
