# Peer review of "The Immunohistochemical Prognostic Value of Nuclear and Cytoplasmic Silent Information Regulator 1 Protein Expression in Saudi Patients with Breast Cancer"

_biomolecules, 2025, doi:10.3390/biom15010050_

Round 1

Reviewer 1 Report

Comments and Suggestions for Authors

The article, “The Immunohistochemical Prognostic Value of Nuclear and Cytoplasmic SIRT1 Protein Expression in Breast Cancer Sub-3 types” by Alhazmi et al., reports the prognostic value of nuclear expression of SIRT1 in breast cancer. However, the presentations are hard to understand for general scientific audience. Please see below.

Major comments

1.      What are the meaning of SIRT1- and SIRT1+ in Table 2, Figure 4, Table 4, Table 5, and Figure 5.

2.      The labels for X- and Y-axis are missing in Figure 3.

Minor comments

1.      Please refer to the official name for SIRT1 in UniProt. The recommended name is NAD-dependent protein deacetylase sirtuin-1 and the alternative names are NAD-dependent protein deacylase sirtuin-1, Regulatory protein SIR2 homolog 1, or SIR2-like protein 1 (hSIR2).

2.      Abbreviations should be explained at their first appearance. For example, LNM in line 31.

3.      Please cite up-to-date reference for Ref. 1.

<The End>

Author Response

We would like to thank the reviewers for their insightful and constructive comments on the manuscript, “The Immunohistochemical Prognostic Value of Nuclear and Cytoplasmic SIRT1 Protein Expression in Breast Cancer Patients.”

Please find the modified manuscript here, where the authors have addressed most of the suggested comments, along with the answers to the reviewers’ comments.

As per your request, the manuscript was also proofread again for any spelling/grammar mistakes and looks now more suitable for publication.

Our point-by-point responses to the comments are detailed in the section below.

Comments and Suggestions for Authors

The article, “The Immunohistochemical Prognostic Value of Nuclear and Cytoplasmic SIRT1 Protein Expression in Breast Cancer Sub-3 types” by Alhazmi et al., reports the prognostic value of nuclear expression of SIRT1 in breast cancer. However, the presentations are hard to understand for general scientific audience. Please see below.

Major comments

  1. What are the meaning of SIRT1- and SIRT1+ in Table 2, Figure 4, Table 4, Table 5, and Figure 5.

  • Authors thank reviewer #1 for his valuable comments. The protein expression patterns are conventionally split into four categories according to the staining intensity: negative (0), weak positive (1+), moderate positive (2+), and strong positive (3+). A light microscope was used at ×40 magnification to estimate the fractions (f0, f1, f2, and f3) of each slide for the four different types of expression intensities (0, 1+,2+,3+) respectively. To quantify the staining results and provide a more comprehensive assessment of both nuclear and cytoplasmic staining, a staining index was calculated. This index was designed to consider not only the intensity of staining observed in the tumor cells but also the proportion/fraction (%) of cells that exhibited each specific level of staining intensity. The formula used for this calculation was originally proposed by Lipponen and Collan (reference: Lipponen PK and Collan Y: Simple quantitation of immunohistochemical staining positivity in microscopy for histopathology routine. Acta Stereologica 1992.) and is expressed as follows:

I = 0 * f0 + 1 * f1 + 2 * f2 + 3 * f3

In this formula: I represent the overall staining index; f0-f3 denote the fractions (%) of cells that exhibit each of the corresponding staining intensity levels (from 0 to 3+) respectively. This calculated index score I ranges from 0 to a maximum of 300. Using the Index score, the overall average score (mean) expression was used as a cutoff point at the nucleus (mean score= 62.43) and cytoplasm (mean score = 30.88). The expression of staining was considered as low expression (SIRT1-) (Index score < mean) and high expression (SIRT1+) (Index score > mean).

  1. The labels for X- and Y-axis are missing in Figure 3

  • The labels for X- and Y-axis have been now added to Figure 3. Thanks

Minor comments

  1. Please refer to the official name for SIRT1 in UniProt. The recommended name is NAD-dependent protein deacetylase sirtuin-1 and the alternative names are NAD-dependent protein deacylase sirtuin-1, Regulatory protein SIR2 homolog 1, or SIR2-like protein 1 (hSIR2).

  • Done, thanks.

  1. Abbreviations should be explained at their first appearance. For example, LNM in line 31.
  • LNM in line 31 stands for (Lymph Node Metastasis). It is now explained at its first appearance.

  1. Please cite up-to-date reference for Ref. 1.

  • As per your request, reference (1) was substituted by this more recent reference:

Siegel, R.L.; Giaquinto, A.N.; Jemal, A. Cancer statistics, 2024. CA: a cancer journal for clinicians 2024, 74(1):12-49. doi:10.3322/caac.21820.

Reviewer 2 Report

Comments and Suggestions for Authors

In the submitted manuscript authors assessed differences in SIRT1 protein expression related to its localization and molecular subtype of breast cancer (BC).

Unfortunately, this manuscript has very many drawbacks, especially related to statistical analyses procedures and presentations of results, but also linguistics, so many things must be corrected and further improved.

1) Lines 33-34: Authors definitively haven't proven that "SIRT1 plays a differential role in sup-pressing or promoting tumor growth depending on the subtype of BC.", they have eventually shown indirect connection!

2) Line 63: "different pivotal cellular pathways and biological processes" must be specified, since such generic statements are worthless.

3) Since it was stressed out that this study is unique for Saudi Arabia, this mast be reflected through the title of this manuscript.

4) Median values must be presented with range, i.e., min.-max., or at least interquartile range. However, since your SIRT1 definitively cannot follow normal distribution, and since the mean value of immunohistoreactivity score is not actual value that you have observed, for dischotomizing SIRT1 into poz./neg. expression you should rather use median value or cutoff should be calculated using AUC-ROC analysis. Furthermore, since even positive SIRT1 expression (>0%) you considered as negative, you cannot claim that "SIRT1 was lost" (line 178). That could be claimed only for cases in which I was actually zero.

5) If you sum cases whose lymph node was analyzed, there were 353 cases (e.g., Table 1, variable "LN"), not 360!!!

6) The tile of figures' panels must be removed, figures properly divided to a, b, c... panels, and panels must be properly defined in each figure legend.

7) All graphs must have title of both x- and y-axes.

8) Preparation of tissue sections and TMA bust be also explained, while besides LOT numbers, also catalog numbers of commercial kits and chemicals must be stated. Also the precise model of used microscope must stated.

9) Creating company of SPSS software must be stated.

10) Survival analysis and its presentation are one of the worst parts of this manuscript! Therefore, endpoints in DSS and DFS must be precisely defined, median and range of follow-up time provided, it must be stated how many patients died and relapsed, independence of prognostic biomarkers must be properly assessed using univariate and multivariate Cox proportional-hazards model regression (e.g., see https://pubmed.ncbi.nlm.nih.gov/21249485/), the complete results of survival analyses must be presented, not just "cherry-picked" significant ones, while results of survival analyses must be also presented with HR and its 95% CI (lower-upper), while x-axis of survival curves should be divided by more meaningful 12 months 8a year), etc.

11) In the main text, always actual p-values, either significant or insignificant, must be provided when results of statistical analyses (comparisons) were presented.

12) It must be stated why 50 years was set as a cutoff value for age.

13) Modified Scarff-Bloom-Richardson grading system (SBR) must be properly referenced.

14) All variables stated in Table 1 must be also listed, and explained, in section "2.1. Patients and samples".

15) All percentages must uniformly be presented with at least one decimal, even whole number percentages, and Table 4 should be re-written that all numbers and text are in just one row.

16) Spearman's correlation coefficient rho must be presented, not r2, and its value must be properly interpreted as in, e.g., https://pmc.ncbi.nlm.nih.gov/articles/PMC3576830/

17) All text from IHC figures presented in Figure 1 must be removed since it is unreadable even after extreme zooming. Figure 1 is not a "summary", eventual presentation of representative cases, while cytoplasmic SIRT1 level 2 seems more intense than level 3! Generally, throughout the text the way how SIRT1 protein expression was presented is constantly mixed and should be unified when authors meant intensity (0-3), score (0-300), or poz./neg., because now it is extremely confusing!

18) Bar graphs on Figures 2 and 4 should be combined that nuclear and cytoplasmic scoring is juxtaposed. Also, in Figure 2 legend it must be explained how data were presented, and this is definitively not a "semi-quantitative scoring", since that is considered only for the scores of protein expression intensity (0-3 "crosses").

19) Figure 3 is a complete mess, and it should be redrawn following my points 11), 12) and 23), i.e., rho and its p-value must be written in graph and sample size stated in figure legend.

20) Results of statistical analyses, i.e., p-values, must be provided also in Tables 2 and 3.

21) Figure 5 is definitively not a "Summary of the distribution of nuclear nuclear (A) and cytoplasmic (B) SIRT1 [what?!]", but eventually summary of statistical analyses or clinico-pathological characteristics that showed a significant association in our breast cancer patients. Also, the maximum value of Y-axis by any chance CANNOT be 150%, so it should be 100%!

22) As reference 1, the newest GLOBOCAN paper (DOI: 10.3322/caac.21834) should be cited and its data presented.

Comments on the Quality of English Language

23) Words "correlated"/"correlation" must be used only in the context of calculated correlation coefficient, otherwise "associated"/"associated" should be used, especially in the context of survival analysis.

24) All full names of genes, proteins, chemicals, etc. abbreviations are not written it title cases, e.g., proper is "sirtuin (SIRT)" for family of genes/proteins, "NAD-dependent protein deacetylase sirtuin-1 (SIRT1)" for protein, "nicotinamide adenine dinucleotide (NAD)", etc.

25) Line 49: SIRT1-SIRT7 are NOT isoforms, but "independent" genes/proteins, members of a gene/protein family!

26) Authors must check that all abbreviations like TN, VI, LNM, H2BC, CI, etc. were explained after their first mentioning in both Abstract and main text, and consistently use them through the manuscript, especially for triple-negative breast cancer.

27) All abbreviations presented in tables and figures must be explained in tables' footnote and figures' legend.

28) Sentence in lines 121-122 is incomprehensible.

29) Lines 95-96: That sentence about conditioning is incomprehensible.

30) The title of section 3.3. is meaningless, as are the statements like "the cytoplasmic SIRT1 didn’t show any significance with ER, PR..." (I suppose you meant "significant association").

31) Sentence in lines 191-192 is incomprehensible, i.e., what means "another way"?!

32) Lines 246-247: It is unclear what authors meant by "tumor invasion" in the statement "most studies investigating the correlation between tumor invasion in BC and SIRT1 expression".

Author Response

We would like to thank the reviewers for their insightful and constructive comments on the manuscript, “The Immunohistochemical Prognostic Value of Nuclear and Cytoplasmic SIRT1 Protein Expression in Breast Cancer Patients.”

Please find the modified manuscript here, where the authors have addressed most of the suggested comments, along with the answers to the reviewers’ comments.

As per your request, the manuscript was also proofread again for any spelling/grammar mistakes and looks now more suitable for publication.

Our point-by-point responses to the comments are detailed in the section below.

Unfortunately, this manuscript has very many drawbacks, especially related to statistical analyses procedures and presentations of results, but also linguistics, so many things must be corrected and further improved.

  • Lines 33-34: Authors definitively haven't proven that "SIRT1 plays a differential role in suppressing or promoting tumor growth depending on the subtype of BC.", they have eventually shown indirect connection!

  • First of all, we would like to sincerely thank reviewer #2 for the insightful comments that have significantly enhanced the quality of this manuscript. Authors TOTALLY AGREE with the reviewer#2 comment regarding the potential differential role of SIRT1 protein expression according to BC subtype. While the cytoplasmic SIRT1 protein overexpression was shown to have a trend of favorable prognostic in the overall BC cohort. However this trend may not stand in all the 4 BC molecular subtypes. On the other side, the no/low nuclear expression of SIRT1 protein was a favorable prognostic factor for the whole cohort.

Authors agree with the reviewer#2 that the analysis of per subtype -where the number of patients per subtype is reduced- can only give hints and “connection” that are worth exploring in future studies using larger cohorts. Therefore, the sentence “SIRT1 plays a differential role in suppressing or promoting tumor growth depending on the subtype of BC." was not accurate and was corrected in the abstract, the discussion and the conclusion.

2) Line 63: "different pivotal cellular pathways and biological processes" must be specified, since such generic statements are worthless.

  • As per your request, the pathways were specified (such as genome integrity, cell growth, cell cycle, and cell death), and included in the manuscript (page 2).

  • Since it was stressed out that this study is unique for Saudi Arabia, this mast be reflected through the title of this manuscript.
  • As per your suggestion, the title was refined as follow: “The Immunohistochemical Prognostic Value of Nuclear and Cytoplasmic SIRT1 Protein Expression in Saudi Breast Cancer Patients”.

  • Median values must be presented with range, i.e., min.-max., or at least interquartile range. However, since your SIRT1 definitively cannot follow normal distribution, and since the mean value of immunohistoreactivity score is not actual value that you have observed, for dischotomizing SIRT1 into poz./neg. expression you should rather use median value or cutoff should be calculated using AUC-ROC analysis. Furthermore, since even positive SIRT1 expression (>0%) you considered as negative, you cannot claim that "SIRT1 was lost" (line 178). That could be claimed only for cases in which I was actually zero.
  • Authors agree with the reviewer’s comment that since the distribution of SIRT1 cannot follow normal distribution, the median values and quartile ranges would be more representative of the cut-offs. Moreover, and as highlighted for reviwer#1, the expression of staining (SIRT1-) means low SIRT1 expression (Index score < mean) and not negative expression, while (SIRT1+) means a high expression of SIRT1 (Index score > mean). Therefore the “SIRT1 was lost” expression was not accurate and therefore corrected by “SIRT1 was low”. The expression (SIRT1-) and (SIRT1+) expressions were further explained whenever required to be more accurate and reflect this latter meaning.

  • If you sum cases whose lymph node was analyzed, there were 353 cases (e.g., Table 2, variable "LN"), not 360!!!
  • We are sorry, this was an addition mistake! The numbers are now fixed in the table1 to show that the correct LN number is 360.

  • The tile of figures' panels must be removed, figures properly divided to a, b, c... panels, and panels must be properly defined in each figure legend.
  • As per your request, the tile of figures' panels or subfigures were removed and replaced by the letters A, B, C, and D. Then, the subtitle of each panel was properly defined in the figure legend.

  • All graphs must have title of both x- and y-axes.
  • Also, the X- and Y- axes were clearly specified in all graphs.

8) Preparation of tissue sections and TMA bust be also explained, while besides LOT numbers, also catalog numbers of commercial kits and chemicals must be stated. Also the precise model of used microscope must stated.

  • As per your request, a paragraph giving details about TMA construction and sectioning was added (page 2, line 87). Moreover, LOT numbers, also catalog numbers of commercial kits and chemicals were also added in the Material & Methods section. The model of the microscope was also specified.

  • Creating company of SPSS software must be stated.
  • Done as per your request.

10) Survival analysis and its presentation are one of the worst parts of this manuscript! Therefore, endpoints in DSS and DFS must be precisely defined, median and range of follow-up time provided, it must be stated how many patients died and relapsed, independence of prognostic biomarkers must be properly assessed using univariate and multivariate Cox proportional-hazards model regression (e.g., see https://pubmed.ncbi.nlm.nih.gov/21249485/), the complete results of survival analyses must be presented, not just "cherry-picked" significant ones, while results of survival analyses must be also presented with HR and its 95% CI (lower-upper), while x-axis of survival curves should be divided by more meaningful 12 months 8a year), etc.

  • We agree with reviewer #2 that the survival presentation and analysis were not well written. Consequently, we have prepared the survival curves once again following your suggestions. Moreover, both DSS and DFS were defined as per your request in the statistical analysis subsection of the manuscript Methods. We have also included 4 survival curves for the whole cohort and rewritten the whole results section (section 3.5) related to the correlation between SIRT1 protein expression patterns and BC patients’ outcomes in order to make these results easily understandable for the readers.

11) In the main text, always actual p-values, either significant or insignificant, must be provided when results of statistical analyses (comparisons) were presented.

  • Authors agree with these comments, and more light was shed on the non significant p-values in the text.

12) It must be stated why 50 years was set as a cutoff value for age.

  • In fact, 50 years was the average of the overall cohort of BC.

13) Modified Scarff-Bloom-Richardson grading system (SBR) must be properly referenced.

13- Done.

14) All variables stated in Table 1 must be also listed, and explained, in section "2.1. Patients and samples".

14- Done.

15) All percentages must uniformly be presented with at least one decimal, even whole number percentages, and Table 4 should be re-written that all numbers and text are in just one row.

15- Done.

16) and 17) Spearman's correlation coefficient rho must be presented, not r2, and its value must be properly interpreted as in, e.g., https://pmc.ncbi.nlm.nih.gov/articles/PMC3576830/. All text from IHC figures presented in Figure 1 must be removed since it is unreadable even after extreme zooming. Figure 1 is not a "summary", eventual presentation of representative cases, while cytoplasmic SIRT1 level 2 seems more intense than level 3! Generally, throughout the text the way how SIRT1 protein expression was presented is constantly mixed and should be unified when authors meant intensity (0-3), score (0-300), or poz./neg., because now it is extremely confusing!

16 and 17- As per your suggestion, the figure was improved. Moreover, we have unified all the intensity of expression levels to avoid confusion. We have also added a detailed description in the results section.

Comments:18), 19), 20), and 21):

18) Bar graphs on Figures 2 and 4 should be combined that nuclear and cytoplasmic scoring is juxtaposed. Also, in Figure 2 legend it must be explained how data were presented, and this is definitively not a "semi-quantitative scoring", since that is considered only for the scores of protein expression intensity (0-3 "crosses").

19) Figure 3 is a complete mess, and it should be redrawn following my points 11), 12) and 23), i.e., rho and its p-value must be written in graph and sample size stated in figure legend.

20) Results of statistical analyses, i.e., p-values, must be provided also in Tables 2 and 3.

21) Figure 5 is definitively not a "Summary of the distribution of nuclear nuclear (A) and cytoplasmic (B) SIRT1 [what?!]", but eventually summary of statistical analyses or clinico-pathological characteristics that showed a significant association in our breast cancer patients. Also, the maximum value of Y-axis by any chance CANNOT be 150%, so it should be 100%!

18, 19, 20, and 21- As per your suggestion:

  • All the figures and tables were revised and improved.
  • The text was removed from the figures and replaced by letters.
  • Figures were split into panels A,B,C, and D whenever required.
  • P-values were double-checked and clearly displayed on the figures.
  • Tables were revised and percentages adjusted and unified in terms of number of decimals
  • As per your request, nuclear and cytoplasmic SIRT expression levels per BC subtype were combined (figure 4).
  • Figures and tables’ legend were revised whenever necessary.
  • Other minor edits were also fixed.

22) As reference 1, the newest GLOBOCAN paper (DOI: 10.3322/caac.21834) should be cited and its data presented.

22- Done.

Comments on the Quality of English Language

23) Words "correlated"/"correlation" must be used only in the context of calculated correlation coefficient, otherwise "associated"/"associated" should be used, especially in the context of survival analysis.

23- Done! Thanks.

24) All full names of genes, proteins, chemicals, etc. abbreviations are not written it title cases, e.g., proper is "sirtuin (SIRT)" for family of genes/proteins, "NAD-dependent protein deacetylase sirtuin-1 (SIRT1)" for protein, "nicotinamide adenine dinucleotide (NAD)", etc.

24- The name of the key genes were double checked. Thanks.

25) Line 49: SIRT1-SIRT7 are NOT isoforms, but "independent" genes/proteins, members of a gene/protein family!

25- Our apologies for this mistake. SIRT are indeed independent genes/proteins, members of a gene/protein family.

26) Authors must check that all abbreviations like TN, VI, LNM, H2BC, CI, etc. were explained after their first mentioning in both Abstract and main text, and consistently use them through the manuscript, especially for triple-negative breast cancer.

26- Done! Thanks.

27) All abbreviations presented in tables and figures must be explained in tables' footnote and figures' legend.

27- Done

28) Sentence in lines 121-122 is incomprehensible.

28- This sentence was rewritten again to be more concise and comprehensible!

29) Lines 95-96: That sentence about conditioning is incomprehensible.

 29- Authors understand the reviewer comment, but we would like to highlight that the immunohistochemistry Ventana used in this study is working through an automated close system that is done through an automated protocol and a set of automatically loaded fluidics that includes mainly the following buffer: EZ Prep™, Cell Conditioning buffer (CC1), UltraView Universal DAB Detection Kit (Lot. No. F30427), which includes DAB Inhibitor and hematoxylin II.  The description of the steps in the methods is just to guide the reader and does not mean a manual protocol. That’s why it was clearly mentioned at the beginning of section (2.3.1) that it is an automated protocol.

30) The title of section 3.3. is meaningless, as are the statements like "the cytoplasmic SIRT1 didn’t show any significance with ER, PR..." (I suppose you meant "significant association").

30- Both the section title and the sentence were improved to be more concise and comprehensible!

31) Sentence in lines 191-192 is incomprehensible, i.e., what means "another way"?!

31- This sentence was rewritten again to be more concise and comprehensible!

32) Lines 246-247: It is unclear what authors meant by "tumor invasion" in the statement "most studies investigating the correlation between tumor invasion in BC and SIRT1 expression".

32- The “tumor invasion” expression in this sentence means advanced stages or metastatic BC. The sentence was improved to showcase this meaning.

Round 2

Reviewer 1 Report

Comments and Suggestions for Authors

Authors have addressed all comments accordingly.

Reviewer 2 Report

Comments and Suggestions for Authors

Authors have satisfactorily responded to my concerns.